# Cytotoxic Activity of α-Aminophosphonic Derivatives Coming from the Tandem Kabachnik–Fields Reaction and Acylation

**DOI:** 10.3390/ph16040506

**Published:** 2023-03-28

**Authors:** Petra R. Varga, Rita Oláhné Szabó, György Dormán, Szilvia Bősze, György Keglevich

**Affiliations:** 1Department of Organic Chemistry and Technology, Faculty of Chemical Technology and Biotechnology, Budapest University of Technology and Economics, 1521 Budapest, Hungary; 2ELKH-ELTE Research Group of Peptide Chemistry, Eötvös Loránd Research Network (ELKH), Eötvös Loránd University (ELTE), 1117 Budapest, Hungary; 3Department of Genetics, Cell and Immunobiology, Semmelweis University, 1089 Budapest, Hungary; 4TargetEx Biosciences, Ltd., 2120 Dunakeszi, Hungary

**Keywords:** α-aminophosphonates, phosphonoylmethyl-α-aminophosphonates, phosphinoylmethyl-α-aminophosphonates, cytotoxic activity

## Abstract

Encouraged by the significant cytotoxic activity of simple α-aminophosphonates, a molecular library comprising phosphonoylmethyl- and phosphinoylmethyl-α-aminophosphonates, a tris derivative, and *N*-acylated species was established. The promising aminophosphonate derivatives were subjected to a comparative structure–activity analysis. We evaluated 12 new aminophosphonate derivatives on tumor cell cultures of different tissue origins (skin, lung, breast, and prostate). Several derivatives showed pronounced, even selective cytostatic effects. According to IC_50_ values, phosphinoylmethyl-aminophosphonate derivative **2e** elicited a significant cytostatic effect on breast adenocarcinoma cells, but it was even more effective against prostatic carcinoma cells. Based on our data, these new compounds exhibited promising antitumor activity on different tumor types, and they might represent a new group of alternative chemotherapeutic agents.

## 1. Introduction

Within organophosphorus compounds [1,2], α-aminophosphonic derivatives are of special importance due to their real and potential biological activity. This is not surprising if the structural analogy between α-aminoalkylphosphonic acids and α-aminocarboxylic acids is considered [3,4,5]. The α-aminophosphonic acid derivatives possess different properties, e.g., they may inhibit enzymes and GABA-receptors, and they may be anti-metabolites [6]. As a consequence, the compounds under discussion may reveal anticancer [7], antibiotic [6], antiviral and HIV [8], anti-inflammatory [9], antimalarial [10], antiasthma [11], antidiabetic [12], as well as antihypertensive effects [13]. Diaryl α-aminophosphonate derivatives are selective and highly potent inhibitors of serine proteases and, hence, can mediate the patho–physical processes of cancer growth, metastasis, osteoarthritis, or heart failure [14].

Various α-aminophosphonate derivatives were investigated and identified as antiproliferative and/or potential anticancer agents. While the diaryl or dialkyl phosphonate ester unit remained unchanged, in most of the cases, different bioactive moieties, such as coumarin [15] or peptidomimetic structures [16], were linked to the amino group. Alternatively, the secondary carbon atom was substituted with heterocycles [17] or nucleobases [18]. In addition, the introduction of trifluoromethyl groups into the arylamino group increased the antiproliferative and apoptosis-inducing properties [19].

The α-aminophopshonic derivatives may be prepared by the Kabachnik–Fields condensation of primary or secondary amines, oxo compounds, such as aldehyde or ketones, and dialkyl phosphites or related derivatives [20,21,22,23]. The three-component reaction has been an excellent model for green chemical studies [24,25,26,27,28]. A modification of the Kabachnik–Fields condensation is the bis(phospha-Mannich) reaction when a primary amine interacts with two equivalents of an aldehyde and a similar quantity of the >P(O)H reactant to afford the corresponding bis derivatives [29,30,31,32]. A special variation is when an α-aminophosphonic derivative is reacted further with another aldehyde and >P(O)H reagent to furnish a phosphonoylmethyl-α-aminophosphonate or a phosphinoylmethyl-α-aminophosphonate [33].

In this article, we wish to report the cytotoxic activity of our products prepared using the tandem Kabachnik–Fields reaction [33]. Even the starting α-aminophosphonates revealed considerable anticancer effects [34].

## 2. Results and Discussion

### 2.1. Synthesis of α-Aminophosphonate Derivatives

The α-benzylamino-benzylphosphonate derivatives (**1a**–**c**) obtained in the Kabachnik–Fields reaction of substituted benzaldehydes, benzylamine, and diethyl phosphite [33] were reacted further in another phospha-Mannich condensation applying paraformaldehyde and diethyl phosphite or secondary phosphine oxide to afford phosphonoylmethyl- or phosphinoylmethyl-benzylamino-benzylphosphonates (**2a**–**f**) (Figure 1) [33]. Compound **1d** (X = MeO) was also prepared from anisic aldehyde but was not reacted further. Aminophosphonate 1e was obtained using aniline instead of benzylamine in condensation with benzaldehyde and diethyl phosphite.

A debenzylation of benzylaminophosphonate **1a** afforded aminophosphonate 3 [33] that was converted to acylated derivatives **4a** and **4b** (Figure 2) [33].

A bis(α,α’-phosphonoylbenzyl)amine (**5**) was prepared starting from benzaldehyde, ammonium acetate, and diethyl phosphite (Figure 3) [33].

α-Amino-benzylphosphonate **3** was a suitable starting material also for tris(phosphonoylmethyl)amine derivative **6** (Figure 4) [33].

The compounds subjected to bioactivity tests are shown in Figure 1.

### 2.2. In Vitro Cytostatic Effect of α-Aminophosphonic Derivatives on Human Tumor Cell Lines

From among the above synthesized α-aminophosphonate derivatives, the cytotoxic activity of *N*-benzyl- and *N*-phenyl α-aminophopshonates **1b**–**e**, phosphonoyl-α-aminophosphonates **2a** and **2b**, phopshinoyl-α-aminophosphonates **2d**–**f**, as well as α-(acylamino)phosphonates **4a** and **4b**, along with bis(diethylphosphonoylphenylmethyl)amine **5** and tris derivative **6** were subjected to investigations on different cancer cells.

To provide structure–activity relations, the in vitro cytostatic activity was evaluated on four human cell lines of different origins: MDA-MB-231 human breast adenocarcinoma [35], A431 human epidermoid carcinoma [36], PC-3 human prostate adenocarcinoma [37], and Ebc-1 human lung squamous cell carcinoma [38].

Dose dependence of the cytostatic effect was studied by treating the cells with the compounds for 24 h, and after removing the active agents, the cells were cultured for another 72 h. The cytostatic effect was expressed in the percentage of the untreated control (Table 1 and Table 2). Compound **6** was insoluble in even DMSO and the aqueous medium applied; therefore, it was excluded from in vitro experiments.

We found that at c = 50 µM, compounds **2a** and **2b** elicited the most pronounced cytostatic effect on the A431 cell line (cytostasis = 43.7 ± 3.5% and 45.1 ± 6.9%, respectively) (Table 2). Compound **2f** proved to be the most cytostatic on both A431 (92.8 ± 0.3%) and Ebc-1 cells (88.5 ± 0.3%). At c = 250 µM, the compound that killed the cells to the greatest extent was **2f** (cytostasis = 90.6 ± 0.3% on MDA-MB 231 cells, 92.8 ± 0.3% on A431 cells, 87.3 ± 0.7% on PC-3 cells, and 90.4 ± 0.3%on Ebc-1 cells at c = 250 um). In general, we can say that A431 human epidermoid carcinoma and Ebc-1 human lung squamous cell carcinoma cells were the most sensitive to the treatment, as most of the compounds already showed a significant cytostatic effect on these cells at a concentration of 50 µM (Table 1 and Table 2).

For those compounds where the cytostatic effect exceeded 50% in certain concentrations, the IC_50_ value that was characteristic of the compound on the appropriate cell line was also determined (Table 3). The cell line on which most of the compounds showed a cytostatic effect over 50% was MDA-MB 231 human breast carcinoma. Considering the IC_50_ values, the most effective was phosphinoylmethyl-aminophosphonate **2e** on MDA-MB 231 cells (IC_50_ = 55.1 µM) and PC-3 cells (IC_50_ = 29.4 µM); its phenyl analog **2d** on MDA-MB 231 cells (IC_50_ = 45.8 µM), and phosphonoylmethyl-aminophosphonate **2b** on A431 cell line (IC_50_ = 53.2 µM). The determined IC_50_ values were also below 100 µM in the case of the following >P(O)CH_2_-aminophosphonates: **2f** (on MDA-MB 231 and Ebc-1 cells), **2b** (on PC-3 cells), and **2a** (on A431 cells).

We investigated the in vitro cytostatic effect of 12 novel α-aminophosphonic derivatives on four human tumor cell lines of different tissue origins (skin, lung, breast, and prostate). To compare the in vitro activity, we employed the end-point type MTT (3-(4,5-dimethylthiazol-2-yl)-2,5-diphenyltetrazolium bromide) assay [39,40,41]. We determined concentration-dependent and cell-selective effects. At a lower concentration (c = 50 µM), Ebc-1 human lung squamous cell carcinoma and A431 human epidermoid carcinoma cells proved to be more sensitive, whereas, in higher concentration (c = 250 µM), MDA-MB-231 human breast adenocarcinoma and PC-3 human prostate adenocarcinoma cells were inhibited to the greatest extent. According to IC_50_ data, phosphinoylmethyl-aminophosphonate derivative **2e** elicited a significant cytostatic effect on MDA-MB 231 breast adenocarcinoma cell line, but it was slightly even more effective against PC-3 cells. **2e** Compound and its phenyl analog (**2d**) showed almost the same cytostatic effect on MDA-MB 231 cells. Compared to the previous studies [42,43,44,45,46], IC_50_ values of those derivatives obtained from the 72 h cytotoxicity data moved in the same range (22.9–352.9 µM) as the cytostasis data we determined after a 24 h treatment and a subsequent 72 h culturing (29.4–169.2 µM).

It is a challenge for us to continue exploring the cytotoxic effect of newer α-aminophospohonates, especially those of optically active derivatives. In order to be able to fulfill this plan, suitable resolution methods have to be elaborated. So far, we have been successful in separating racemic α-hydroxyphosphonates into their enantiomers [47].

### 2.3. In Silico Target Assessment of the Major Cytotoxic Hit Compounds

Since the mid-2000s, searchable annotated molecular databases have been available that contain biological activities, frequently together with the protein targets they act on [48].

BindingDB [49] (https://www.bindingdb.org, accessed on 31 January 2023), currently containing 107,154 active compounds with biological data, gives the opportunity to predict the protein targets of compounds found active in cellular assays based on the “similar structure—similar property” principle. Some recent examples from the literature where similarity search was used for initial target assessment are in [50,51,52]. We can identify structurally similar compounds that interact with numerous protein targets, and further analysis would reveal the putative protein targets of cell-based hit compounds.

The similarity search on the best aminophosphonate cytotoxic hit compound (**2b**, **2d**, **2e**) was carried out using InstantJChem software (ChemAxon, Budapest, Hungary) (Figure 2), which uses 2D molecular fingerprints for comparison. Compounds were considered similar if the Tanimoto coefficient was ≥0.5. Although 0.5 represents a moderate similarity threshold, it provides a broad preliminary target pool for more specific and detailed analysis. Of course, in vitro biological screening could only validate the prediction.

The number of bioactivity data with putative targets is, all together: 71 (**2b**: 58, **2d**/**2e**: 25). (The lower number is due to the overlap of findings) (Table 4).

The resulting putative protein targets can be analyzed and attempted to link to disease states represented by the cell lines (e.g., MDA-MB-231 human breast, PC-3 prostate, and A431 skin cancer) where the best aminophosphonate cytotoxic hits were identified. Thus, at first, the predicted targets were classified as either cancer or non-cancer-associated targets. Secondly, the cancer-related targets were further investigated to determine whether the cell lines express or even overexpress such proteins and whether their overexpression could promote the progression of the tumor. All the identified cancer-related targets meet the specific OncoScore criteria and are listed among the cancer-associated genes (see Supplementary Materials of Ref. [53]). The small molecules that inhibit any of such targets could contribute to decreasing tumor growth as the aminophosphonate hits expressed cytotoxic/antiproliferative effects on the cancer cell lines.

The similar compounds related to our cytotoxic hits, together with their annotated protein targets, are shown in Table 5.

Short description of the putative targets.

Furin

Proprotein convertases, including furin, have been implicated in the activation of a wide variety of various biological pathways and could facilitate tumor formation and progression. The expression of furin has been confirmed in various cancers such as head and neck cell carcinoma, breast cancer, etc. [53].

Several peptidomimetic and related small molecule inhibitors have been identified over the years. In general, phosphates and phosphonates could often mimic the peptide bonds; thus, they are replaceable in peptidomimetic inhibitor design.

Prostatic Acid Phosphatase

Prostatic acid phosphatase (PAP) originates in the prostate and is normally present in small amounts in the blood. Elevated levels of PAP in patients indicate the progression of various tumors such as prostate cancer [55], testicular cancer, leukemia, etc. Various α-benzylaminobenzylphosphonic acid derivatives have been developed over the years using structure-based drug design methods that led to the identification of low nanomolar inhibitors [56].

Tyrosine Phosphatase Enzyme Family

The protein tyrosine phosphatase enzyme family (PTPs) removes phosphate groups from proteins. PTPs are considered potential drug targets against several diseases, including cancer [57], obesity, diabetes, etc. [58]. Several PTPs overexpressed in human cancers could act as targets (oncogene) or “anti-targets” in tumor therapy since they could also suppress tumor progression upon overexpression [59].

Based on recent data, the most important anticancer target is PTPN11 (SHP2). Overexpression of SHP2 is associated with an increased risk of leukemia [60], breast cancer [61], and skin cancer cell lines [62]. Several PTPN11 (SHP2) inhibitors have been developed so far, with an increasing focus on selectivity toward the anti-target PTPs [63]. Most of the 2nd generation compounds lack P-containing moieties; however, their structure partially mimics the transition state complex of the enzyme–substrate interaction.

In summary, based on the above in silico target prediction, we could conclude that our hit compounds that showed cytotoxicity against breast, prostate, and skin cancer cell lines might act through the above-described protein targets by applying the similarity principle. Detailed molecular dynamics and docking studies and, most importantly, target-based bioassays would confirm the above prediction. These future efforts are out of the scope of the present paper.

## 3. Materials and Methods

### 3.1. Synthesis of α-Aminophosphonate Derivatives

The phosphonoylmethyl- and phosphinoylmethyl-benzylamino-benzylphosphonates **2a**–**f**, acylated derivatives **4a** and **4b**, along with bis- and tris derivatives **5** and **6**, respectively, were synthesized as described earlier [32].

### 3.2. Cell Lines and Culture Conditions—In Vitro Cytotoxicity Assays of Carcinoma Cell Lines

In vitro cytostatic effect of the compounds was determined on MDA-MB-231 human breast adenocarcinoma [36], A431 human epidermoid carcinoma [37], PC-3 human prostate adenocarcinoma [38], and Ebc-1 human lung squamous cell carcinoma [39] cell lines. Cells were cultured in DMEM medium supplemented with 10% FBS, 2 mM L-glutamine, 50 IU/mL penicillin/50 μg/mL streptomycin antibiotic cocktail, 1 mM sodium pyruvate, and 1% non-essential amino acid mixture at 37 °C in a humidified atmosphere with 5% CO_2_. At the confluent state, cells were plated into 96-well tissue culture plates with the initial cell number of 5.0 × 10^3^ cells/well. Cells were treated with the compounds after 24 h in serum-free medium containing 1.0 *v*/*v*% DMSO at 2, 10, 50, and 250 μM concentrations. Control cells were treated with serum-free medium only or with DMSO (c = 1.0 *v*/*v* %) at the same conditions. After overnight incubation, cells were washed twice with a serum-free medium; then, they were cultured for another 72 h in complete culturing medium at 37 °C. Following that, MTT-solution (at c = 0.37 mg/mL final concentration) was added to each well. The respiratory chain [40] and other electron transport systems [41] reduce MTT and, thereby, form non-water-soluble violet formazan crystals within the cell [42]. After 3 h of incubation with MTT, the cells were centrifuged for 5 min at 900× *g*, and then the supernatant was removed. The obtained formazan crystals were dissolved in DMSO (100 µL), and the optical density (OD) of the samples was measured at λ = 540 nm and 620 nm, respectively, using ELISA Reader (iEMS Reader, Labsystems, Vantaa, Finland). OD_620_ values were subtracted from OD_540_ values. The amount of these crystals serves as an estimate for the number of mitochondria and, hence, the number of living cells in the well [43]. The percent of cytostasis was calculated with the following equation:Cytostatic effect (%) = [1−(OD_treated_/OD_control_)] × 100
where values OD_treated_ and OD_control_ corresponds to the optical densities of the treated and the control wells, respectively. In each case, two independent experiments were carried out with 4 parallel measurements; statistical analysis of data was performed using Student’s *t*-test at the 95% confidence level. The 50% inhibitory concentration (IC_50_) values were determined from the dose–response curves, which were defined using Microcal™ Origin 2018 software: cytostasis was plotted as a function of concentration, on which a sigmoidal curve was fitted. Based on this curve, the half-maximal inhibitory concentration (IC_50_) value was determined, which was expressed in micromolar units.

## 4. Conclusions

We have designed, synthesized, and in vitro evaluated 12 novel aminophosphonate derivatives, the majority of which comprised species with two P-functions. The data from cytostatic activity on skin, lung, breast, and prostate tissue-originated tumor cultures revealed that these compounds could be classified as new α-aminophosphonic derivatives with promising antitumor properties due to the pronounced and selective cytostatic effect. Our results are consistent with the observations of the past two decades, which proved that some α-aminophosphonic derivatives could have antitumor and genotoxic effects on different tumor cell cultures in vitro. According to IC_50_ values, phosphinoylmethyl-aminophosphonate **2e** elicited a remarkable cytostatic effect on breast adenocarcinoma and on prostatic carcinoma cells. Moreover, our study revealed that Ebc-1, MDA-MB231, and A431 carcinoma cultures were generally sensitive to the treatments with the compounds, and there was significant inhibition detected at the 50–100 μM concentration range.

In silico target prediction proposed that the identified cytotoxic compounds would act through the interaction with furin, prostatic acid phosphatase, and tyrosine phosphatase protein targets that are highly expressed in the tumor cell lines used in the present study.

## Data Availability

Data is contained within the article.

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
