# Peer review of "Cytotoxic Activity of α-Aminophosphonic Derivatives Coming from the Tandem Kabachnik–Fields Reaction and Acylation"

_pharmaceuticals, 2023, doi:10.3390/ph16040506_

Round 1

Reviewer 1 Report

Account

The presented work of Petra R. Varga, Rita Oláhné Szabó, György Dormán, Szilvia BÅ‘sze? György Keglevich is devoted to the Cytotoxic Activity of α-Aminophosphonic Derivatives Coming from the Tandem Kabachnik–Fields Reaction. Among organophosphorus compounds, α-aminophosphonic derivatives are of particular importance in connection with their real and potential biological activity. it is not surprising if there is a structural analogy between α-aminoalkylphosphonic acids and α-aminocarboxylic acids. Literature data prove that various α-aminophosphone derivatives can have a wide variety of biological activities. The antitumor effect of some derivatives of α-aminophosphones was studied in vitro and in vivo on four human leukemic cell lines. The in vitro cytostatic effect of several acylated dialkyl-α-hydroxy-benzylphosphonates has been found to exceed 30% in several human tumor cell lines. The authors synthesized and evaluated in vitro 12 new derivatives of aminophosphonates. Data on cytostatic activity in tumor cultures of the skin, lungs, breast, and prostate showed that these compounds can be classified as new α-aminophosphonic derivatives with promising antitumor properties due to a pronounced and selective cytostatic effect. It was found that phosphinoylmethyl-aminophosphonates cause a pronounced cytostatic effect on mammary adenocarcinoma and prostate carcinoma cells. The synthesis of aminophosphonate derivatives is simple and requires the use of microwaves. These new compounds show promising antitumor activity against various types of tumors and may represent a new group of alternative chemotherapeutic agents.

The work is well done, the conclusions made by the authors are convincing. In this work, the main results are summarized in tables. There is a large bibliography of 53 references.

The presented work will undoubtedly be a successful stage in the development of new anticancer agents. I believe that the work will be interesting and useful for specialists interesting by antitumor compounds. The work can be published in the Journal “Pharmaceuticals”

Author Response

We are grateful for this Referee for the positive comments and for approving the ms.

Reviewer 2 Report

The authors examined the cytotoxicity of several aminophosphonic derivatives and conducted in silico search to justify the structural effects. The novelty and scope look fine and the results are solid. However, organization of this manuscript needs to be improved. My suggestions see below:

Firstly, the focus of this manuscript is the effect and mechanisms of the aminophosphonic derivatives rather than their synthesis. However, the authors pay too much attention on the synthesis. 1. In introduction they nearly spend half of the context for the synthesis, which is not worth that many words. They should briefly mention the methods and focus more on the how/why such compounds are cytotoxic by presenting more examples (e.g. the docking between similar molecules and enzyme pockets as inhibitor). 2. I suggest the authors move the synthetic methodologies from the result and discussion section to the experimental section. In the result and discussion section the authors can start with presenting the library and discussion on why they are selected/designed. 3. In the last sentence of the abstract the authors emphasize two facts of this reaction but without further presenting their ideas on the facts. This does not align with the story and should be removed. 

Secondly, the introduction and the conclusion are weak. 1. Contents between line 267-274 should be discussed in the introduction part since this provides some background information. 2. In the introduction part the authors should presents some concrete examples of the cytotoxicity of organophosphorus compounds and their mechanisms like a mini-review in few paragraphs. 

Thirdly, it would be very helpful if the authors can 1. make some simple molecular dynamics or DFT-based 3D modeling of molecules (the calculation isn’t intensive with some small basis sets) and 2. Conduct chiral resolution of some compound in the library that are found with high cytostatic effect. The 2D structures are very different from 3D structures and simply analyzing from the molecular connectivity is not cogent enough. The authors used “similar structure – similar property” principle (by the way, there should be some citations on successful/classical examples or high quality reviews in line 161 and 224 for this topic, which are missing) and compare these oganophosphorous compunds with peptide bonds. They should further discuss on the 3D structures including the chirality to support their findings in table 5.

Lastly, some typos should be fixed. E.g. line 47 “Mannch” and line 91 “was evaluated on on four human cell lines…”. This is not meant to be a thorough grammar/typo check and the authors should proofread twice. 

Author Response

Replies to Referee 2:

 „The authors examined the cytotoxicity of several aminophosphonic derivatives and conducted in silico search to justify the structural effects. The novelty and scope look fine and the results are solid. However, organization of this manuscript needs to be improved. My suggestions see below:

 Firstly, the focus of this manuscript is the effect and mechanisms of the aminophosphonic derivatives rather than their synthesis. However, the authors pay too much attention on the synthesis. 1. In introduction they nearly spend half of the context for the synthesis, which is not worth that many words. They should briefly mention the methods and focus more on the how/why such compounds are cytotoxic by presenting more examples (e.g. the docking between similar molecules and enzyme pockets as inhibitor).”

- The request was obeyed: on the one hand, keeping the original references, the synthetic information was shortened (concentrated). On the other hand, more knowledge was provided on the cytotoxic activity of alfa-aminohosphonates. Typical structural modifications were mentioned. See new references [15-19].

„2. I suggest the authors move the synthetic methodologies from the result and discussion section to the experimental section. In the result and discussion section the authors can start with presenting the library and discussion on why they are selected/designed.”

- This wish was not followed, as synthetic details were not provided due to the fact that the just published article was cited (ref. [32]). At the same time, an informative sentence on this situation was inserted at the beginning of the Experimental as 3.1.

„3. In the last sentence of the abstract the authors emphasize two facts of this reaction, but without further presenting their ideas on the facts. This does not align with the story and should be removed.” 

- The last sentence of the abstract was deleted.

„Secondly, the introduction and the conclusion are weak. 1. Contents between line 267-274 should be discussed in the introduction part since this provides some background information.”

- The part outlined by lines 267-274 in the Conclusions was deleted. This information is available in the earlier parts. We have well expanded the discussion of the bioactivities. The details regarding IC50 values etc. can be read in the section of Results and Discussion.

„2. In the introduction part the authors should presents some concrete examples of the cytotoxicity of organophosphorus compounds and their mechanisms like a mini-review in few paragraphs.” 

- The information on the cytotoxicity of the aminophosphonates in general was brodened by adding new references 15-19, as mentioned above.

„Thirdly, it would be very helpful if the authors can 1. make some simple molecular dynamics or DFT-based 3D modeling of molecules (the calculation isn’t intensive with some small basis sets)”

- The following statement was included: „Detailed molecular dynamics and docking studies and most importantly target-based bioassays would confirm the above prediction. These future efforts are out of the scope of the present paper.”

„and 2. Conduct chiral resolution of some compound in the library that are found with high cytostatic effect. The 2D structures are very different from 3D structures and simply analyzing from the molecular connectivity is not cogent enough.”

- We included the following text: „It is a challenge for us to continue exploring the cytotoxic effect of newer α-aminophospohonates, especially those of optically active derivatives. In order to be able to fulfill this plan, suitable resolution methods have to be elaborated. So far, we have been succesful in separating racemic α-hydroxyphosphonates into their enantiomers [46].”

„The authors used “similar structure – similar property” principle (by the way, there should be some citations on successful/classical examples or high quality reviews in line 161 and 224 for this topic, which are missing) and compare these oganophosphorous compunds with peptide bonds.”

- The principle was supported by inserting three new references [49-51].

„They should further discuss on the 3D structures including the chirality to support their findings in table 5.”

- There were some uncertainities regarding the asymmetric centres in Table 5. This uncertain information has now been removed. The Table was provided with a remark „the annotated bioactivity databases often lack the chirality information”.

„Lastly, some typos should be fixed. E.g. line 47 “Mannch” and line 91 “was evaluated on on four human cell lines…”. This is not meant to be a thorough grammar/typo check and the authors should proofread twice.” 

- These mistakes were all corretcted, and the ms was double-checked.

For the corrections, see the yellow highlights!

Thanks for the constructive remarks!